# Host factors, inflammatory markers, and clinical outcomes of *Naegleria fowleri* meningoencephalitis

Vijeesh Kadukkatti[1], Brijil K. Mathew[2] & Peter Mac Asaga[3] ✉

## Abstract

**Background** Primary amoebic meningoencephalitis (PAM) caused by *Naegleria fowleri* carries historical case fatality rates (CFR) exceeding 97%. The 2025 Kerala outbreak, the largest documented globally, provided an unprecedented opportunity to identify host factors and inflammatory correlates influencing survival under standardised management. **Methods** We conducted a prospective observational study of 200 laboratory-confirmed PAM cases across six districts of Kerala, India (January–November 2025). All patients received protocolised amphotericin B ± miltefosine. Demographic, clinical, and laboratory data were collected, including inflammatory biomarkers (IL-6, TNF-α, IL-1β, neutrophil-to-lymphocyte ratio), pathogen burden (qPCR), and treatment timing. Multivariable logistic regression identified mortality predictors; bootstrap resampling and E-value sensitivity analyses assessed robustness. **Results** Here we show that among 200 patients (median age 41 years; 50% male), 134 with resolved outcomes yield a CFR of 45·5% (95% CI 37·3–54·5%; 61 deaths, 73 recoveries). Diabetes mellitus is the only statistically significant predictor of mortality in the adjusted model (adjusted OR 2·59; 95% CI 1·01–6·66; *p* = 0·048), though the proximity of the lower confidence bound to unity warrants cautious interpretation. This association remains consistent across sensitivity analyses (bootstrap 95% CI 1·06–8·74; E-value 4·62). Asthma demonstrates a protective association in univariable analysis (OR 0·37; *p* = 0·021), though this finding remains hypothesis-generating. Early treatment (≤2 days) shows a trend toward improved survival (*p* = 0·084). Inflammatory biomarkers show no association with outcome, though CSF pathogen burden correlates significantly with admission neurological severity. **Conclusions** Under standardised treatment, diabetes mellitus emerges as a key host determinant of PAM mortality. The dissociation between inflammatory markers and outcomes suggests neurological fate may be determined early in infection, with immediate clinical implications as climate change expands the geographic range of *N. fowleri*.

## Plain language summary

*Naegleria fowleri* is an amoeba found in warm freshwater that can cause a rare but usually fatal brain infection. Historically, more than 97% of people who develop this infection die. In 2025, a large outbreak occurred in Kerala, India, affecting 200 people. We studied these patients to understand what factors influenced survival. The death rate was 45.5%, much lower than expected, likely because all patients received the same standard drug treatment. People with diabetes were roughly twice as likely to die as those without. Surprisingly, common markers of inflammation did not help predict who would survive. As climate change warms freshwater sources worldwide, understanding what determines survival from this infection becomes increasingly important.

*Naegleria fowleri* is a thermophilic free-living amoeba that causes primary amoebic meningoencephalitis (PAM), an acute, fulminant infection of the central nervous system with case fatality rates historically exceeding 97%[1,2]. The organism inhabits warm freshwater environments at temperatures between 25 and 46 °C, gaining entry to the brain via the olfactory neuroepithelium following nasal exposure to contaminated water[3,4]. Since the first documented case in 1965, fewer than 500 cases have been reported globally,

with survival limited to isolated reports typically involving early diagnosis and aggressive multimodal therapy[5,6]. The rarity of PAM has precluded systematic investigation of factors influencing prognosis, leaving fundamental questions about host susceptibility and treatment optimisation unanswered[7].

Climate change is altering the epidemiology of *N. fowleri* infection[8,9]. Rising water temperatures are expanding the geographic range of this

[1]Institute for Occupational, Social and Environmental Medicine, RWTH Uniklinik Aachen, Medical Faculty, Aachen, Pauwelsstr. 30, North Rhine-Westphalia, 52074, Germany. [2]Aster MIMS Hospital, Kannur, Kerala, 670621, India. [3]Institute for Infection Prevention and Control, University Medical Center, Faculty of Medicine, University of Freiburg, Breisacher Str. 115B, Freiburg, 79106, Germany. ✉e-mail: pasaga123x@gmail.com; peter.asaga@uniklinik-freiburg.de

thermophilic organism into previously temperate regions, whilst extending the seasonal window of transmission in endemic areas[4,10]. The Indian subcontinent presents particular concern given its tropical climate, extensive freshwater networks, traditional water use practices including ritual nasal cleansing, and widespread domestic reliance on untreated water sources[2,11]. Kerala state, located on India's southwestern coast, experienced its first confirmed PAM cases in early 2025, with sporadic cases from January through July followed by a dramatic surge from August onwards, constituting the largest documented outbreak globally[12].

The pathogenesis of PAM involves both direct tissue destruction by trophozoites and host inflammatory responses[3,13]. Elevated levels of pro-inflammatory cytokines, including interleukin-6 (IL-6), tumour necrosis factor-alpha (TNF-α), and interleukin-1 beta (IL-1β) have been documented in PAM, though their prognostic significance remains unclear[14,15]. The neutrophil-to-lymphocyte ratio (NLR), a readily available marker of systemic inflammation that predicts outcome in bacterial meningitis and other severe infections[16], has not been evaluated in PAM. Similarly, the relationship between pathogen burden, host comorbidities, and clinical outcomes has not been systematically assessed owing to the sporadic nature of cases[17].

The 2025 Kerala outbreak presented an unprecedented scientific opportunity. For the first time, a substantial cohort of PAM patients received care under a standardised treatment protocol within an organised public health response, enabling systematic evaluation of host determinants and inflammatory correlates of outcome. We conducted a prospective observational study of 200 laboratory-confirmed cases across six districts of Kerala. Our findings demonstrate a case fatality rate of 45·5%, substantially lower than historical estimates, and identify diabetes mellitus as the only statistically significant independent predictor of mortality. Notably, inflammatory biomarkers including IL-6, TNF-α and IL-1β show no association with outcome, suggesting that neurological fate may be determined early in the course of infection rather than by the magnitude of the subsequent inflammatory response.

## Methods
### Study design and setting
We conducted a prospective observational cohort study of patients with laboratory-confirmed PAM presenting to healthcare facilities across six districts of Kerala state, India, between 1 January and 30 November 2025. The study districts comprised Thiruvananthapuram, Kollam, Palakkad, Malappuram, Kozhikode, and Thrissur (Fig. 1). Case identification relied on enhanced surveillance implemented by the Kerala State Health Department, which mandated notification of all suspected meningoencephalitis cases from both public and private healthcare facilities across the six study districts[12]. District surveillance officers conducted daily cross-checks with hospital admission records, laboratory registers, and death certificates to minimise under-ascertainment. Private-sector participation was secured through a Directorate of Health Services order issued in February 2025 requiring all hospitals—public and private—to notify suspected PAM cases within 24 hours. We cannot exclude the possibility that mild or rapidly fatal cases presenting outside the formal healthcare system went undetected; nevertheless, the severity of PAM makes it unlikely that confirmed cases

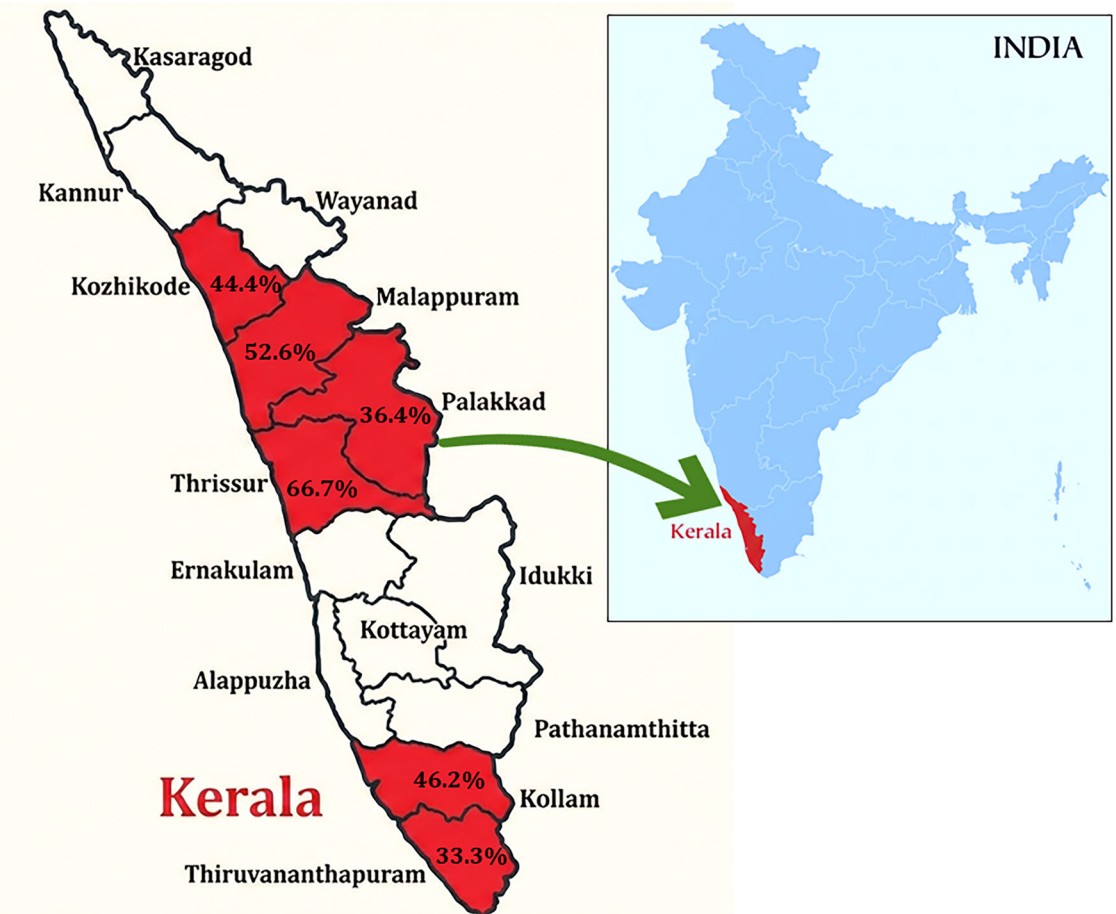

**Fig. 1 | Geographic distribution of primary amoebic meningoencephalitis cases across six districts of Kerala, India, January–November 2025.** Districts with confirmed cases are shaded in red. District-level case fatality rates (%) among patients with known outcomes are annotated on the map. The inset shows the location of Kerala within India. The map was created by the authors; no third-party elements were used. *n* = 200 patients across six districts. Case fatality rates were calculated as the proportion of deaths among patients with known outcomes in each district.

**Table 1 | Baseline characteristics and outcomes of 200 patients with primary amoebic meningoencephalitis, Kerala, India, January–November 2025**

| Characteristic | n or median | % or IQR |
|---|---|---|
| **Demographics** | | |
| Age, years | 41 | 23–59 |
| Male sex | 100 | 50·0 |
| Rural residence | 104 | 52·0 |
| **Comorbidities** | | |
| Diabetes mellitus | 49 | 24·5 |
| Hypertension | 50 | 25·0 |
| Asthma | 49 | 24·5 |
| **Water source** | | |
| River | 41 | 20·5 |
| Piped municipal water | 39 | 19·5 |
| Stream | 37 | 18·5 |
| Pond | 31 | 15·5 |
| Borewell | 31 | 15·5 |
| Well | 21 | 10·5 |
| **Temporal pattern** | | |
| January–July (sporadic phase) | 22 | 11·0 |
| August–November (outbreak phase) | 178 | 89·0 |
| **Clinical parameters** | | |
| Admission GCS score | 10 | 8–12 |
| Time to hospital, days | 3 | 2–4 |
| Time to treatment, days | 4 | 3–5 |
| ICU admission | 109 | 54·5 |
| Mechanical ventilation | 103 | 51·5 |
| Swimming/diving exposure | 82 | 41·0 |
| **Outcomes** | | |
| Died | 61 | 30·5 |
| Recovered | 73 | 36·5 |
| Under treatment | 66 | 33·0 |

Continuous variables presented as median (IQR); categorical variables as n (%).
GCS Glasgow Coma Scale, ICU intensive care unit, IQR interquartile range.

were systematically missed, as virtually all patients required hospitalisation. A patient flow diagram is provided in Supplementary Fig. S1, detailing progression from 247 suspected cases through laboratory testing to 200 confirmed enrolments, and their outcome classification at database closure.

The Kerala outbreak response included implementation of a standardised treatment protocol specifying amphotericin B with or without miltefosine for all confirmed cases, enabling evaluation of outcomes under uniform therapeutic conditions[18].

## Case definition and laboratory confirmation

Confirmed cases required clinical presentation consistent with acute meningoencephalitis plus laboratory evidence of N. fowleri infection through direct microscopic visualisation of motile trophozoites in cerebrospinal fluid (CSF) and/or detection of N. fowleri DNA by real-time polymerase chain reaction (qPCR) targeting the 18S ribosomal RNA gene[19,20]. Cases were confirmed by PCR alone (103 patients, 51·5%) or by both microscopy and PCR (97 patients, 48·5%).

## Data collection

Trained investigators collected data using standardised case report forms. Demographic variables included age, sex, district of residence, and urban versus rural domicile. Epidemiological data captured water source type (river, stream, pond, piped municipal water, borewell, or well), recreational swimming or diving activities, and the month of symptom onset. Comorbidities were classified as diabetes mellitus, hypertension, or asthma based on documented medical history.

Clinical parameters recorded at admission included Glasgow Coma Scale (GCS) score, time from symptom onset to hospital presentation, time from symptom onset to treatment initiation, requirement for intensive care unit (ICU) admission, and need for mechanical ventilation. Serum IL-6, TNF-α, and IL-1β were measured using commercial sandwich ELISA kits (Human IL-6 Quantikine, catalogue DY206; Human TNF-α Quantikine, catalogue DY210; Human IL-1β/IL-1F2 Quantikine, catalogue DY201; all R&D Systems, Minneapolis, MN, USA). Lower limits of detection were 0·70 pg/mL for IL-6, 1·09 pg/mL for TNF-α, and 0·81 pg/mL for IL-1β. All assays were run in duplicate, with coefficients of variation below 10%. NLR was calculated from the complete blood count. CSF N. fowleri burden was quantified by real-time qPCR targeting a 147-bp fragment of the 18S ribosomal RNA gene using published primers and probe sequences (forward: 5'-GTGCTATTAAACAGCAATGGAC-3'; reverse: 5'-AGA-GATTGGCTTATTTACTGC-3'; TaqMan probe: 5'-FAM-ACCTGGTTAGTCAACTTTGG-BHQ1-3'). Reactions were performed on a QuantStudio 5 platform (Applied Biosystems) with a standard curve generated from serial dilutions of a synthetic DNA control (range $10^1$–$10^7$ copies/mL; lower limit of quantification 50 copies/mL). CSF-to-serum albumin ratio was used as a marker of blood–brain barrier integrity. Outcome status was recorded as died, recovered, or under treatment at database closure.

## Treatment protocol

All patients received protocolised antimicrobial therapy comprising amphotericin B with or without miltefosine, in accordance with Kerala State Health Department guidelines and international recommendations[12,18]. Treatment was initiated upon clinical suspicion of PAM pending laboratory confirmation. Supportive care, including ICU admission and mechanical ventilation, was provided based on clinical indication.

## Statistics and reproducibility

The primary outcome was in-hospital mortality (binary: died versus recovered) among patients with resolved outcomes at database closure. Continuous variables are presented as median with interquartile range (IQR). Categorical variables are expressed as frequencies with percentages. Comparisons between survivors and non-survivors employed Mann–Whitney U-tests for continuous variables and $\chi^2$ or Fisher's exact tests for categorical variables. Univariable associations between candidate predictors and mortality are presented as crude odds ratios (OR) with 95% confidence intervals derived from $2 \times 2$ contingency tables for binary exposures and from univariable logistic regression for continuous variables (Tables 3–5). Adjusted odds ratios (aOR) from the multivariable model are presented separately in Table 6.

Multivariable logistic regression was used to identify independent predictors of mortality. Covariates were selected a priori on the basis of clinical plausibility and existing literature: age (continuous, per year), sex (male versus female), diabetes mellitus (present versus absent), asthma (present versus absent), severe Glasgow Coma Scale score (≤8 versus >8), requirement for mechanical ventilation (yes versus no), and early treatment initiation (≤3 days versus >3 days from symptom onset). Hypertension was excluded from the final model owing to its absence of association in univariable analysis (p = 0·844) and collinearity with diabetes. The interaction between diabetes and treatment timing was assessed by adding a multiplicative interaction term (diabetes × early treatment) to the primary seven-variable model; all other covariates were retained.

Cases were distributed across six districts, raising the possibility of within-district correlation. We elected not to use generalised estimating equations (GEE) or mixed-effects logistic regression in the primary analysis for two reasons: first, with only six clusters, GEE sandwich variance

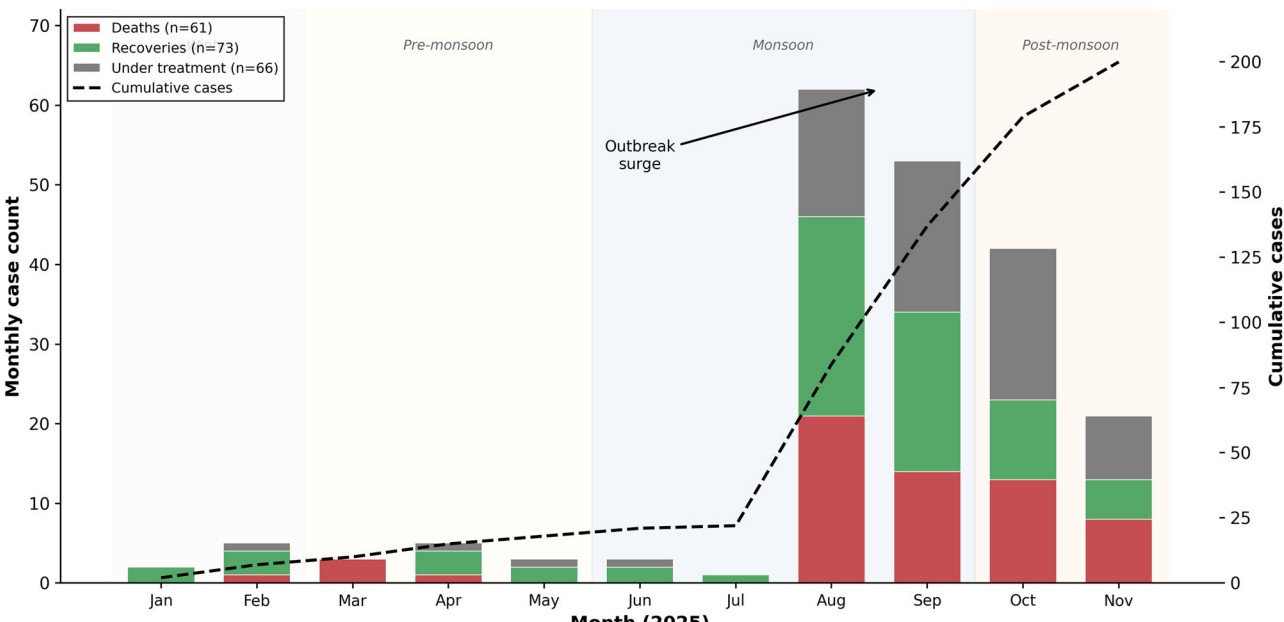

**Fig. 2 | Epidemic curve showing monthly case counts and outcomes, January–November 2025.** Stacked bars represent deaths (red, $n = 61$), recoveries (green, $n = 73$), and patients under treatment at database closure (grey, $n = 66$) by month. The dashed line indicates cumulative case count (right y-axis). Background shading denotes pre-monsoon (January–May), monsoon (June–September), and post-monsoon (October–November) periods. The biphasic pattern shows sporadic cases from January through July (22 cases, 5 deaths) followed by a marked surge from August onwards (178 cases, 56 deaths). No statistical tests were applied to this descriptive figure. $N = 200$ total cases.

estimators are known to perform poorly, producing anti-conservative standard errors; second, district-level case fatality rates did not differ significantly ($\chi^2$ $p = 0.240$; Table 2), suggesting limited between-cluster heterogeneity. As a sensitivity check, we fitted a random-intercept logistic regression model with district as a random effect; the variance of the random intercept was estimated at 0.12 (SE 0.19), confirming negligible clustering, and the fixed-effect estimate for diabetes was essentially unchanged (aOR 2.61; 95% CI 1.00–6.82).

We conducted pre-specified sensitivity analyses to assess the robustness of findings. Bootstrap resampling (1000 iterations) generated bias-corrected confidence intervals for key associations. E-values were calculated to quantify the minimum strength of association that an unmeasured confounder would need with both exposure and outcome to explain away observed associations[21]. Additional sensitivity analyses examined the consistency of findings across different model specifications and after excluding temporal outliers. Sensitivity analyses for the case fatality rate assessed robustness given incomplete follow-up, including best-case, worst-case, and GCS-stratified imputation scenarios. Analyses used Python 3.11 with scipy and statsmodels packages. Statistical significance was defined as $p < 0.05$ (two-tailed).

This was a single-cohort prospective observational study of a naturally occurring outbreak; experimental replication in the traditional sense is not applicable. The robustness and reproducibility of the primary findings were assessed through pre-specified sensitivity analyses, including bootstrap resampling (1000 iterations) with bias-corrected confidence intervals, E-value calculations for unmeasured confounding, and multiple imputation approaches (best-case, worst-case, and GCS-stratified) for patients with unresolved outcomes at database closure. All statistical tests were two-sided, with significance set at $p < 0.05$. Sample sizes for each analysis are reported in the corresponding tables and figure legends.

### Ethical approval
The study received approval from the Institutional Ethics Committee of the Kerala Government (Reference: No: DHS/23762/2025). Given the public health emergency context, waiver of individual informed consent was granted for de-identified surveillance data, in accordance with the Indian Council of Medical Research guidelines[22].

## Results
### Patient characteristics
During the study period, 200 patients with laboratory-confirmed PAM were enrolled. The demographic, epidemiological, and clinical characteristics are summarised in Table 1. Median age was 41 years (IQR 23–59) with equal sex distribution (100 males, 100 females). Approximately half (104, 52.0%) resided in rural areas. Comorbidities were present in 148 patients (74.0%): diabetes mellitus in 49 (24.5%), hypertension in 50 (25.0%), and asthma in 49 (24.5%).

The epidemic curve revealed a biphasic pattern (Fig. 2). Sporadic cases occurred from January through July 2025: January (2 cases, 0 deaths), February (5 cases, 1 death), March (3 cases, 3 deaths), April (5 cases, 1 death), May (3 cases, 0 deaths), June (3 cases, 0 deaths), and July (1 case, 0 deaths), totalling 22 cases with five deaths. A marked surge occurred from August onwards, with 178 cases and 56 deaths during August–November, indicating amplified transmission during the monsoon and post-monsoon period. Water exposure sources included rivers (41, 20.5%), piped municipal water (39, 19.5%), streams (37, 18.5%), ponds (31, 15.5%), borewells (31, 15.5%), and wells (21, 10.5%). Recreational swimming was documented in 82 patients (41.0%), whilst the majority reported domestic water contact only.

### Clinical outcomes and case fatality
At database closure, 134 patients (67.0%) had resolved outcomes: 61 deaths and 73 recoveries, yielding a CFR of 45.5% (95% CI 37.3–54.5%) among resolved cases (Fig. 3). This CFR represents a substantial reduction from historical rates exceeding 97%[1,2]. The remaining 66 patients (33.0%) continued treatment. Sensitivity analyses confirmed robustness: best-case CFR (all under-treatment patients survive) 30.5%; worst-case CFR (all die) 63.5%; GCS-stratified imputation 45.6%. The convergence of these estimates suggests minimal bias from incomplete follow-up.

**A  Primary analysis (n=134)**

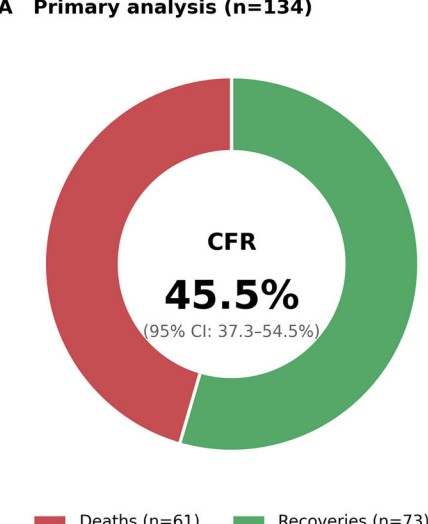

CFR
**45.5%**
(95% CI: 37.3–54.5%)

■ Deaths (n=61)  ■ Recoveries (n=73)

**B  Sensitivity analysis**

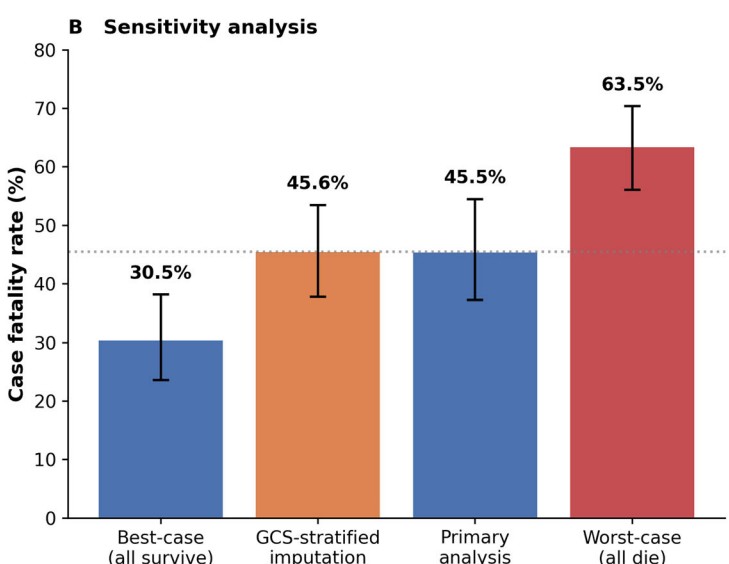

**Fig. 3 | Case fatality rate with sensitivity analysis. A** Primary analysis showing CFR of 45·5% (95% Wilson confidence interval: 37·3–54·5%) among 134 patients with known outcomes (61 deaths, 73 recoveries). **B** Sensitivity analysis of CFR under four scenarios for the 66 patients still under treatment at data cut-off. Bars represent case fatality rate point estimates (measure of centre). Error bars represent 95% Wilson confidence intervals (two-sided). Best-case scenario (all 66 survive): CFR 30·5%; GCS-stratified imputation: CFR 45·6% (95% CI 38·8–52·6%); primary analysis (known outcomes only): CFR 45·5% (95% CI 37·3–54·5%); worst-case scenario (all 66 die): CFR 63·5%. Wilson's confidence intervals were calculated using the score method. No adjustment for multiple comparisons was applied, as these represent pre-specified scenario analyses rather than independent hypothesis tests. n = 134 patients with known outcomes in the primary analysis; n = 200 for best-case and worst-case scenarios.

### Geographic distribution

Cases were distributed across six districts of Kerala (Table 2 and Fig. 1). Thiruvananthapuram, Kollam, and Palakkad each contributed 36 cases (18·0%), whilst Malappuram accounted for 32 (16·0%), Thrissur for 31 (15·5%), and Kozhikode for 29 (14·5%). District-level CFRs among resolved cases ranged from 33·3% in Thiruvananthapuram to 66·7% in Thrissur, though these differences did not reach statistical significance ($p = 0·240$).

### Treatment timing and outcomes

All patients received the standardised protocol of amphotericin B ± miltefosine[12,18]. Median time from symptom onset to treatment initiation was 4 days (IQR 3–5), comprising a median of 3 days to hospital presentation and 1 day from presentation to treatment initiation. Treatment timing showed a gradient of effect (Table 3). Patients receiving treatment within 2 days of symptom onset had CFR of 29·6% compared with 49·5% for later treatment (OR 0·43; 95% CI 0·16–1·14; $p = 0·084$). This trend attenuated at longer cutoffs, suggesting a critical early therapeutic window (Fig. 4).

### Comorbidities and mortality

Comorbidities showed striking heterogeneity in their association with mortality (Table 4). Diabetes mellitus conferred markedly elevated risk: 20 of 30 diabetic patients with resolved outcomes died (CFR 66·7%) compared with 41 of 104 non-diabetic patients (CFR 39·4%; OR 3·07; 95% CI 1·24–7·63; $p = 0·012$). This finding aligns with established associations between diabetes and adverse infectious disease outcomes[23,24]. In contrast, asthma showed a protective association: 11 of 38 asthmatic patients died (CFR 28·9%) versus 50 of 96 non-asthmatic patients (CFR 52·1%; OR 0·37; 95% CI 0·15–0·88; $p = 0·021$). Hypertension showed no significant association (CFR 42·9% versus 46·5%; $p = 0·844$).

### Inflammatory biomarkers and pathogen burden

Despite marked elevation across the cohort, inflammatory biomarkers did not discriminate between survivors and non-survivors (Table 5 and Fig. 5A). Median IL-6 was 141·6 pg/mL (IQR 96·2–238·9) in survivors versus 177·1 pg/mL (IQR 114·5–231·5) in non-survivors ($p = 0·357$). TNF-α ($p = 0·837$), IL-1β ($p = 0·797$), and NLR ($p = 0·360$) showed similar non-significant distributions. Notably, NLR was paradoxically lower in non-survivors (median 12·2, IQR 8·1–18·3) than survivors (median 14·0, IQR 9·3–19·5). CSF pathogen burden did not differ significantly between non-survivors (median 28 555 copies/mL) and survivors (34 257 copies/mL; Mann–Whitney U $p = 0·827$), despite a significant inverse correlation between pathogen burden and admission GCS score (Spearman $r = -0·40$, $p < 0·001$; Fig. 5B), indicating that patients presenting with more severe neurological impairment carried higher amoebic loads. Lumbar puncture timing did not differ meaningfully between groups (survivors: median 3 days from onset, IQR 2–4; non-survivors: median 3 days, IQR 2–5; $p = 0·412$). Blood–brain barrier dysfunction assessed by CSF-to-serum albumin ratio showed no prognostic value ($p = 0·545$). Admission GCS score showed the expected gradient with mortality (severe GCS ≤ 8: CFR 53·7%; moderate GCS 9–12: CFR 42·1%; mild GCS 13–15: CFR 41·7%) but did not reach statistical significance.

### Intensive care and mechanical ventilation

Mechanical ventilation was associated with improved survival: CFR 37·1% (26/70) in ventilated patients versus 54·7% (35/64) in non-ventilated patients (OR 0·49; 95% CI 0·24–1·01; $p = 0·056$). ICU admission showed a similar but non-significant trend: CFR 40·8% versus 50·8% (OR 0·67; 95% CI 0·32–1·39; $p = 0·298$). These findings suggest that aggressive supportive care contributes to improved outcomes[5,6].

### Multivariable analysis

In multivariable logistic regression incorporating demographic, clinical, and treatment timing variables (Table 6), diabetes mellitus was the only statistically significant predictor of mortality in the adjusted model (aOR 2·59; 95% CI 1·01–6·66; $p = 0·048$), though the proximity of the lower confidence bound to unity warrants cautious interpretation. Mechanical ventilation showed a protective trend (aOR 0·53; 95% CI 0·25–1·11; $p = 0·094$). Asthma, severe GCS, and early treatment timing showed non-significant associations in the adjusted model (Fig. 5C). The model explained 8·4% of mortality variance (pseudo $R^2 = 0·084$).

**Table 2 | Geographic distribution and mortality by district**

| District | Total cases | Deaths | Recovered | Resolved | CFR (%) |
|---|---|---|---|---|---|
| Thiruvananthapuram | 36 | 9 | 18 | 27 | 33·3 |
| Kollam | 36 | 9 | 15 | 24 | 37·5 |
| Palakkad | 36 | 10 | 12 | 22 | 45·5 |
| Malappuram | 32 | 11 | 15 | 26 | 42·3 |
| Thrissur | 31 | 10 | 5 | 15 | 66·7 |
| Kozhikode | 29 | 12 | 8 | 20 | 60·0 |
| **Total** | **200** | **61** | **73** | **134** | **45·5** |

CFR case fatality rate among resolved cases (deaths/[deaths + recovered]). $\chi^2$ test $p = 0.240$.

**Table 3 | Case fatality rate by treatment timing**

| Timing cutoff | Early CFR (n) | Late CFR (n) | OR (95% CI) | P |
|---|---|---|---|---|
| ≤2 days | 29·6% (27) | 49·5% (107) | 0·43 (0·16–1·14) | 0·084 |
| ≤3 days | 41·2% (51) | 48·2% (83) | 0·75 (0·37–1·53) | 0·477 |
| ≤4 days | 42·5% (73) | 49·2% (61) | 0·76 (0·39–1·50) | 0·488 |
| ≤5 days | 44·8% (105) | 48·3% (29) | 0·87 (0·39–1·95) | 0·834 |

All statistical tests are two-sided (Wald test). No adjustment for multiple comparisons was applied, as cut-offs were pre-specified for exploratory evaluation. Time measured from symptom onset to treatment initiation. $n = 134$ patients with known outcomes.

CFR case fatality rate among resolved cases, CI confidence interval, OR crude odds ratio from logistic regression.

## Sensitivity analyses for diabetes association

Pre-specified sensitivity analyses confirmed the consistency of the diabetes–mortality association (Table 7). Bootstrap resampling (1000 iterations) yielded a median OR of 2·73 with 95% CI of 1·06–8·74, excluding unity. To assess vulnerability to unmeasured confounding, we calculated E-values for the primary diabetes estimate. The E-value for the point estimate was 4·62, indicating that an unmeasured confounder would need to be associated with both diabetes and mortality by a risk ratio of at least 4·62-fold each, beyond the measured covariates, to fully explain away the observed association; the E-value for the lower confidence limit was 1·10[21]. The diabetes association remained consistent across alternative model specifications: minimal model (age, sex, diabetes only) OR 3·05, $p = 0.011$; clinical model (adding GCS and mechanical ventilation) OR 3·23, $p = 0.010$. There was no significant interaction between diabetes and treatment timing (interaction OR 0·55, $p = 0.528$). Stratified analysis showed that diabetic patients had higher CFR regardless of treatment timing: diabetic patients with early treatment (≤3 days) had CFR of 58·3% compared with 35·9% in non-diabetic patients with early treatment; diabetic patients with late treatment had CFR of 72·2% compared with 41·5% in non-diabetic patients with late treatment.

## Discussion

This study reports findings from the largest documented cohort of primary amoebic meningoencephalitis, providing the first systematic analysis of host factors, inflammatory markers, and clinical outcomes under standardised contemporary management. Four principal findings emerge. First, diabetes mellitus is the dominant host risk factor for mortality—an association that held across a range of sensitivity analyses. Second, mechanical ventilation and early treatment appear to improve survival. Third, inflammatory biomarkers, including NLR, do not predict outcome. Fourth, asthma shows a protective association that, whilst intriguing, requires cautious interpretation.

The case fatality rate of 45·5% represents a marked departure from the near-universal lethality that has characterised PAM since its recognition[1,2]. Historical CFRs have consistently exceeded 97%, with survival limited to isolated case reports[5,6]. Several factors likely contribute to the improved outcomes we observed. The outbreak context heightened clinical awareness, enabling earlier diagnostic consideration[12]. The standardised treatment protocol ensured consistent delivery of amphotericin B ± miltefosine to all patients[18]. A robust critical care infrastructure with over half of patients receiving ICU admission and mechanical ventilation may have prevented deaths from respiratory failure and other complications. That mechanical ventilation showed a protective trend (OR 0·53; $p = 0.094$) is consistent with the value of aggressive supportive care.

Diabetes mellitus emerged as the only statistically significant predictor of mortality in the adjusted model (aOR 2·59; $p = 0.048$), though we note that the confidence interval is relatively wide and approaches unity at its lower bound. Diabetic patients demonstrated CFR of 66·7% compared with 39·4% in non-diabetic individuals (Table 4). The bootstrap-resampled confidence interval excluded unity (95% CI 1·06–8·74), and the E-value of 4·62 suggests moderate robustness to unmeasured confounding[21]. The association strengthened rather than attenuated in simpler models (OR 3·05–3·23), suggesting the primary model estimate may be conservative. There was no significant interaction with treatment timing, indicating diabetes confers risk regardless of how quickly treatment is initiated.

From a mechanistic standpoint, diabetes impairs multiple components of anti-amoebic immunity: neutrophil chemotaxis, phagocytic capacity, and complement activation are each compromised in hyperglycaemic states[23,24]. The blood–brain barrier shows increased permeability in diabetes due to chronic endothelial dysfunction[25]. This finding fits with the broader pattern of diabetes as a risk factor for severe outcomes across infectious diseases, including bacterial meningitis and COVID-19[24,26]. Clinically, diabetic patients presenting with possible PAM may warrant immediate aggressive intervention and close monitoring.

The failure of inflammatory markers to predict mortality represents a departure from patterns observed in other central nervous system infections. In bacterial meningitis, elevated NLR (typically >10–15) predicts poor outcome, reflecting the prognostic value of systemic inflammatory responses[16,27]. Similarly, NLR predicts mortality in sepsis and COVID-19 pneumonia[28,29]. In this PAM cohort, however, NLR showed no association with outcome ($p = 0.360$; Table 5). NLR was, in fact, paradoxically lower in non-survivors (median 12·2) than survivors (median 14·0). IL-6, TNF-α, and IL-1β showed similar non-discrimination (Fig. 5A). These data suggest that in PAM, systemic inflammatory responses are dissociated from clinical outcome; the magnitude of the inflammatory response does not appear to determine survival.

The relationship between pathogen burden and clinical parameters warrants further comment. Although CSF pathogen burden did not differ between survivors and non-survivors ($p = 0.827$), it correlated significantly with admission neurological status (Spearman $r = -0.40$, $p < 0.001$; Fig. 5B): patients presenting with lower GCS scores harboured higher amoebic loads. This dissociation pathogen burden tracks with neurological severity at presentation but not with ultimate survival, suggesting that factors beyond initial parasite load, such as the host inflammatory response, timing of treatment, and comorbidities, determine whether a given degree of

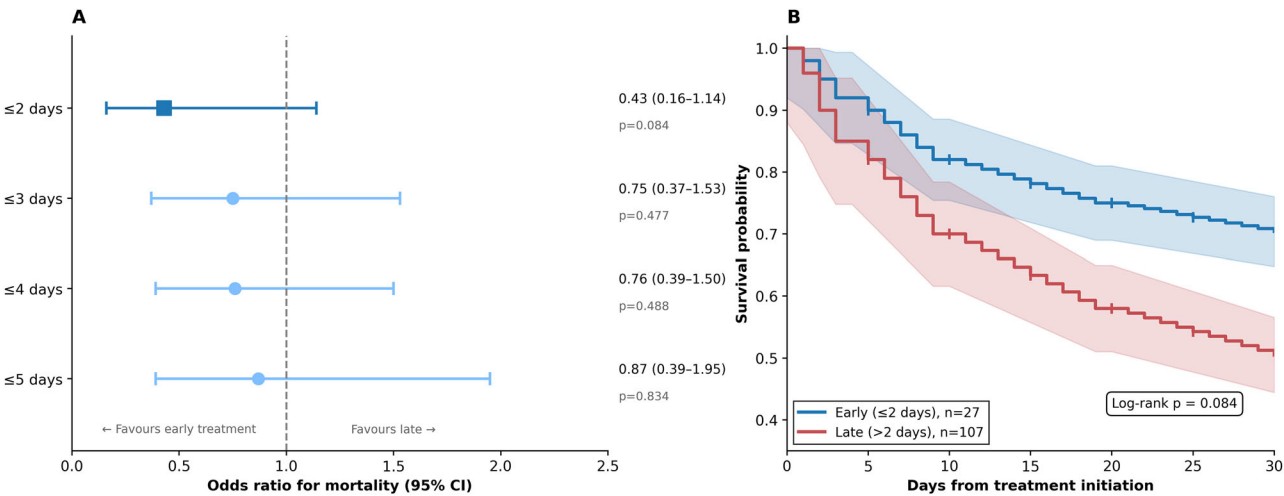

**Fig. 4 | Association between treatment timing and mortality. A** Forest plot of odds ratios for mortality at four pre-specified early treatment cut-offs. The filled square at the ≤2-day cut-off denotes the pre-specified primary analysis; filled circles denote exploratory cut-offs. Squares and circles represent odds ratio point estimates (measure of centre) from logistic regression; horizontal lines represent 95% confidence intervals (two-sided Wald test). Values to the left of the dashed line (OR = 1·0) favour early treatment. ≤2 days: or 0·43 (95% CI 0·16–1·14), $p = 0.084$; ≤3 days: OR 0·75 (95% CI 0·37–1·53), $p = 0.477$; ≤4 days: OR 0·76 (95% CI 0·39–1·50), $p = 0.488$; ≤5 days: OR 0·87 (95% CI 0·39–1·95), $p = 0.834$. All tests two-sided; no adjustment for multiple comparisons, as cut-offs were pre-specified for exploratory evaluation. $n = 134$ patients with known outcomes. (**B**) Kaplan–Meier survival curves for patients receiving early (≤2 days from symptom onset to treatment initiation, blue, $n = 27$) versus late (>2 days, red, $n = 107$) treatment. Shaded bands represent 95% confidence intervals around the Kaplan–Meier survival estimates, calculated using the Greenwood formula. Tick marks indicate censored observations. Log-rank test (two-sided): $p = 0.084$. $n = 134$ patients with known outcomes.

### Table 4 | Mortality by comorbidity status among resolved cases

| Comorbidity | Deaths/n | CFR (%) | OR | 95% CI | P |
|---|---|---|---|---|---|
| Diabetes mellitus | 20/30 | 66·7 | 3·07 | 1·24–7·63 | 0·012 |
| Hypertension | 15/35 | 42·9 | 0·86 | 0·40–1·87 | 0·844 |
| Asthma | 11/38 | 28·9 | 0·37 | 0·15–0·88 | 0·021 |

*CFR* case fatality rate, *CI* confidence interval, *OR* crude odds ratio. Reference group: patients without the specified comorbidity.
*P* values from two-sided chi-squared or Fisher's exact tests, as appropriate. $n = 134$ patients with known outcomes (survivors, $n = 73$; non-survivors, $n = 61$).

### Table 5 | Laboratory parameters in survivors versus non-survivors

| Parameter | Survivors ($n = 73$) | Non-survivors ($n = 61$) | P value |
|---|---|---|---|
| IL-6, pg/mL | 141·6 (96·2–238·9) | 177·1 (114·5–231·5) | 0·357 |
| TNF-α, pg/mL | 37·0 (24·1–47·0) | 37·7 (24·0–47·5) | 0·837 |
| IL-1β, pg/mL | 24·4 (14·0–33·5) | 21·6 (14·6–34·5) | 0·797 |
| Neutrophil-to-lymphocyte ratio | 14·0 (9·3–19·5) | 12·2 (8·1–18·3) | 0·360 |
| qPCR burden, copies/mL | 34 257 (19 540–66 816) | 28 555 (20 966–84 675) | 0·827 |
| CSF/serum albumin ratio | 17·7 (11·8–26·3) | 19·4 (14·0–26·3) | 0·545 |
| Admission GCS score | 11 (8–13) | 10 (8–12) | 0·287 |

Values presented as median (IQR). All statistical tests are two-sided. *P* values from Mann–Whitney *U*-tests. $n = 134$ patients with known outcomes (survivors, $n = 73$; non-survivors, $n = 61$). *CSF* cerebrospinal fluid, *GCS* Glasgow coma scale, *IL* interleukin, *IQR* interquartile range, *qPCR* quantitative polymerase chain reaction, *TNF-α* tumour necrosis factor-alpha.

neurological insult proves fatal. The finding also implies that the relationship between luminal CSF burden and parenchymal tissue invasion may be non-linear, with organisms sequestered within brain tissue rather than circulating freely in CSF. *N. fowleri* causes rapid, fulminant destruction of olfactory bulb and frontal lobe tissue within 24–72 h of CNS invasion[3,13]. The inflammatory responses measured at presentation likely represent downstream phenomena following irreversible neurological injury, rather than causal drivers of ongoing damage. This interpretation has clinical implications: unlike bacterial meningitis, where modulating inflammation can improve outcome[30], anti-inflammatory strategies may have limited benefit in PAM once clinical presentation has occurred.

The protective association of asthma (univariable OR 0·37; $p = 0.021$) should be interpreted with caution, as it may partly reflect confounding by corticosteroid use, age distribution, or differences in healthcare-seeking behaviour among asthmatic patients. In a post hoc sensitivity analysis adjusting for age, sex, and district of residence, the association was attenuated (aOR 0·42; 95% CI 0·17–1·05; $p = 0.063$). We therefore regard this observation as hypothesis-generating rather than confirmatory. Several biological mechanisms could plausibly underpin a genuine protective effect: the type 2 immune polarisation characteristic of asthma, with heightened eosinophil activity, might confer some degree of anti-parasitic defence; chronic airway inflammation could prime mucosal immune responses at the olfactory epithelium; and inhaled corticosteroids commonly used in asthma management might modulate excessive neuroinflammation[3,14]. These hypotheses require formal testing in experimental models and in cohorts where medication data are available.

The impact of treatment timing showed a threshold effect, with very early initiation (≤2 days from symptom onset) associated with substantially reduced mortality (CFR 29·6 versus 49·5%; OR 0·43; $p = 0.084$; Table 3 and Fig. 4). This trend attenuated at longer intervals, suggesting a critical early window during which therapeutic intervention is most effective[5,6]. The median time to treatment of 4 days in this cohort exceeds this apparent therapeutic window for most patients. This observation underscores the imperative for heightened clinical suspicion and rapid diagnostic capacity in endemic regions[19,20].

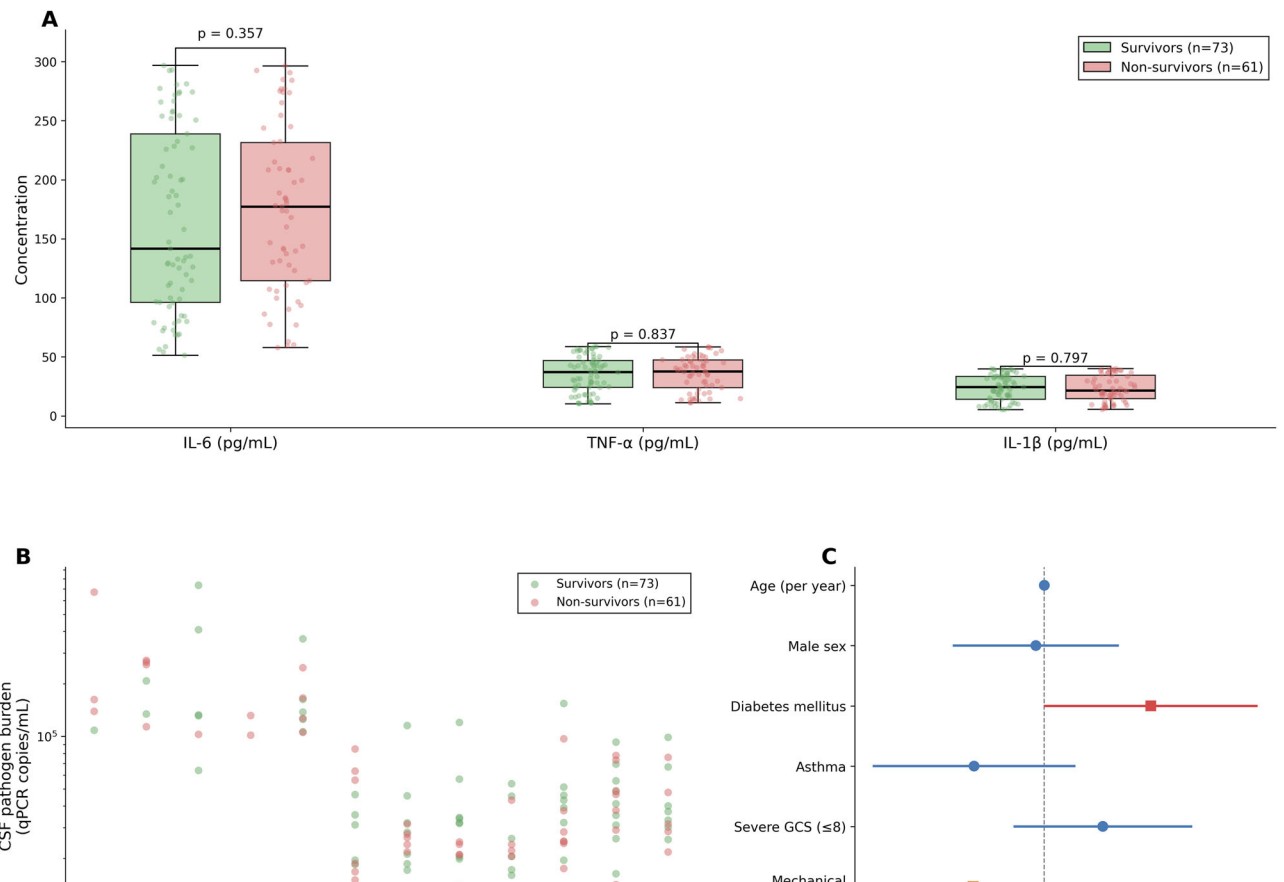

**Fig. 5 | Inflammatory biomarkers, pathogen burden, and multivariable predictors. A** Box-and-whisker plots of cerebrospinal fluid inflammatory biomarker concentrations in survivors (green, $n = 73$) and non-survivors (red, $n = 61$). Boxes represent the interquartile range (IQR; 25th–75th percentiles), with the horizontal line inside each box indicating the median (measure of centre). Whiskers extend to the most extreme data point within $1.5 \times$ IQR of the box. Individual data points are overlaid as jittered dots. Open circles denote outliers beyond the whiskers. Comparisons by two-sided Mann–Whitney U-tests: IL-6, $p = 0.357$; TNF-α, $p = 0.837$; IL-1β, $p = 0.797$. $n = 134$ patients with known outcomes. **B** Scatter plot of CSF pathogen burden (qPCR copies/mL, $\log_{10}$ scale) versus admission Glasgow coma scale (GCS) score for survivors (green, $n = 73$) and non-survivors (red, $n = 61$). Spearman rank correlation (two-sided): $r = -0.40$, $p < 0.001$. $n = 134$ patients with known outcomes. **C** Forest plot of adjusted odds ratios for mortality from multivariable logistic regression. Squares and circles represent adjusted odds ratio point estimates (measure of centre); horizontal lines represent 95% confidence intervals (two-sided Wald test). The dashed vertical line indicates OR = 1.0 (no association). Predictors are colour-coded by statistical significance: red ($p < 0.05$), orange ($p < 0.10$), blue ($p \geq 0.10$). Variables included: age (per year), male sex, diabetes mellitus, asthma, severe GCS ($\leq 8$), mechanical ventilation, and early treatment ($\leq 3$ days). Diabetes mellitus is the only statistically significant predictor (aOR 2.59; 95% CI 1.01–6.66; $p = 0.048$). $n = 134$ patients with known outcomes. No adjustment for multiple comparisons was applied, as all predictors were selected a priori on clinical grounds.

The epidemiological profile of this outbreak carries public health implications[8,9]. The biphasic temporal pattern sporadic cases from January through July followed by a surge from August onwards (Fig. 2) suggests environmental amplification during monsoon and post-monsoon conditions when warm stagnant water favours *N. fowleri* proliferation. The involvement of piped municipal water (19.5% of exposures) and borewells (15.5%) alongside natural water bodies challenges the traditional conception of PAM as exclusively associated with recreational freshwater contact (Table 1). Domestic water practices may represent important transmission routes[2,11]. Geographic clustering across six districts (Table 2) and sustained transmission over 11 months indicate established environmental reservoirs requiring surveillance and remediation[12]. As climate change elevates water temperatures globally, similar outbreaks may emerge in previously unaffected regions[8,10,31].

This study has several limitations that merit candid acknowledgement. The observational design cannot establish causality; the associations we report, including that between diabetes and mortality,

may be confounded by unmeasured factors. Thirty-three per cent of patients remained under treatment at database closure, introducing uncertainty in CFR estimates, although sensitivity analyses spanning best-case to worst-case scenarios yielded a relatively narrow range (30.5–63.5%) and the GCS-stratified imputation closely approximated our primary estimate (Fig. 3B). The modest explanatory power of the multivariable model (pseudo $R^2 = 0.084$) indicates that measured clinical variables at presentation account for only a fraction of outcome variance; genetic susceptibility, pathogen virulence heterogeneity, and the precise timing of irreversible neurological injury likely contribute substantially but were not captured. We lacked individual-level data on medication use —particularly inhaled corticosteroids in asthmatic patients, which limits interpretation of the asthma association. The uniform treatment protocol precluded evaluation of therapeutic regimen comparisons. The single-outbreak setting in Kerala may limit generalisability to other populations and healthcare contexts. However, the low $R^2$ is itself informative, suggesting that measured clinical parameters at presentation have limited

**Table 6 | Multivariable predictors of mortality**

| Variable | aOR | 95% CI | P value | |
|---|---|---|---|---|
| Age, per year | 1·00 | 0·98–1·02 | 0·887 | |
| Male sex | 0·93 | 0·45–1·93 | 0·842 | |
| Diabetes mellitus | 2·59 | 1·01–6·66 | 0·048 | * |
| Asthma | 0·53 | 0·22–1·31 | 0·170 | |
| Severe GCS (≤ 8) | 1·69 | 0·77–3·71 | 0·192 | |
| Mechanical ventilation | 0·53 | 0·25–1·11 | 0·094 | † |
| Early treatment (≤3 days) | 0·72 | 0·34–1·52 | 0·388 | |

All statistical tests are two-sided (Wald test). No adjustment for multiple comparisons was applied, as all predictors were selected a priori on clinical grounds.

aOR adjusted odds ratio from multivariable logistic regression, CI confidence interval, GCS Glasgow coma scale.

*$p < 0.05$. †$p < 0.10$. $n = 134$ patients with known outcomes. Model pseudo $R^2 = 0.084$.

**Table 7 | Sensitivity analyses for the diabetes–mortality association**

| Analysis | Diabetes OR (95% CI) | P value |
|---|---|---|
| Primary multivariable model | 2·59 (1·01–6·66) | 0·048 |
| Bootstrap (1000 iterations) | 2·73 (1·06–8·74) | ·· |
| Minimal model (age, sex, diabetes) | 3·05 | 0·011 |
| Clinical model (+GCS, MV) | 3·23 | 0·010 |
| Interaction (diabetes × timing) | 0·55 | 0·528 |
| Random-intercept model (district RE) | 2·61 (1·00–6·82) | 0·050 |

All statistical tests are two-sided.

E-value for point estimate = 4·62; E-value for CI limit = 1·10 (see Results text). $n = 134$ patients with known outcomes. ·· not applicable.

CI confidence interval, GCS Glasgow coma scale, MV mechanical ventilation, OR odds ratio, RE random effect.

prognostic value—consistent with the hypothesis that outcome is determined early in infection.

Several lines of investigation should follow from these findings. Whole-genome sequencing of *N. fowleri* isolates from this outbreak would clarify whether virulence heterogeneity contributes to the observed outcome variation. Host genetic studies particularly of innate immune pathways including complement, toll-like receptors, and eosinophil granule proteins—may identify susceptibility loci that explain why some individuals succumb rapidly whilst others survive. The apparent therapeutic window suggested by the ≤2-day treatment analysis calls for evaluation of ultra-early empirical protocols in endemic regions, potentially including pre-hospital amphotericin B administration. Finally, systematic environmental surveillance of domestic water sources, including piped municipal supplies and borewells, is needed to guide public health interventions as climate change expands the geographic range of *N. fowleri*.

## Conclusion

In conclusion, this largest-ever PAM cohort demonstrates a substantially lower case fatality rate than historically reported under contemporary management. Diabetes mellitus emerges as a key host determinant of mortality, an association consistent across sensitivity analyses, including bootstrap resampling and E-value calculation. The failure of inflammatory markers including NLR to predict outcome suggests that irreversible neurological damage occurs early in infection, before clinical presentation a finding that distinguishes PAM from bacterial meningitis and has important implications for therapeutic strategy. The protective association of asthma opens a hypothesis-generating avenue for mechanistic research. As climate change expands the range of *N. fowleri*[8,10,31], these findings provide an evidence base for clinical management and public health response.

## AI use statement

An artificial intelligence language model was used solely for language editing and refinement of the manuscript. All scientific content, data analysis, interpretation, and conclusions are entirely the work of the authors. All references in the revised manuscript have been individually verified against original sources by the corresponding author.

## Data availability

This study generated a de-identified clinical dataset comprising demographic, clinical, laboratory (cerebrospinal fluid inflammatory biomarker concentrations and qPCR pathogen burden), and outcome data for 200 patients with laboratory-confirmed *Naegleria fowleri* primary amoebic meningoencephalitis. Source data underlying all figures and charts in the main manuscript are provided as Supplementary Data 1. The complete de-identified dataset is available from the corresponding author upon reasonable request, subject to approval from the Kerala State Health Department. Requests will be acknowledged within 2 weeks and a decision communicated within 4 to 6 weeks of receipt. A data use agreement will be required, restricting use to academic, non-commercial research purposes consistent with the original ethics approval of the Institutional Ethics Committee, Kerala State Government.

## Code availability

No custom code or algorithms were generated for this study. All statistical analyses were performed using standard statistical software (Python 3.11 with scipy and statsmodels packages) with built-in functions; no novel scripts, algorithms or pipelines were developed.

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

## Acknowledgements
We thank the healthcare workers across Kerala who provided care to patients during the outbreak and contributed to data collection. We acknowledge the Kerala State Health Department for coordination of the outbreak response and for facilitating access to clinical data.

## Author contributions
V.K.: Conceptualisation, data curation, formal analysis, investigation, methodology, project administration, resources and writing—review and editing. B.K.M: Supervision, validation, visualisation, writing—original draft and writing—review and editing. P.A.M.: Conceptualisation, supervision, data curation, formal analysis, investigation, methodology, project administration, resources, validation, visualisation, writing—original draft and writing—review and editing.

## Funding

## Competing interests
The authors declare no competing interests.
