## [Transparent Peer Review file · Communications Medicine]

Host factors, inflammatory markers, and clinical outcomes of *Naegleria fowleri* meningoencephalitis.

Corresponding Author: Dr Peter Asaga

Version 0:

Reviewer comments:

Reviewer #1

(Remarks to the Author)

This manuscript presents an important analysis of the largest documented cohort of primary amoebic meningoencephalitis. The study evaluates 200 laboratory confirmed patients from the Kerala 2025 outbreak and investigates host factors, inflammatory markers and treatment timing in relation to survival under a standardised clinical protocol. The main conclusions are that diabetes mellitus is the only independent predictor of mortality, that very early treatment may improve survival, and that inflammatory biomarkers and pathogen burden show no association with outcome. The authors also note a possible protective association with asthma. The observed case fatality rate of 45.5 per cent is significantly lower than historic reports and will be of interest to the field.

The dataset is rare for this disease and the findings are relevant. The identification of diabetes as a key mortality determinant is convincing. The absence of predictive value for IL-6, TNF-alpha, IL-1beta and the neutrophil to lymphocyte ratio challenges established assumptions derived from bacterial meningitis and other central nervous system infections.

Several areas require clarification to strengthen the manuscript. The patient recruitment pathway should be described more clearly. It is important to know precisely how patients were identified, whether any cases presenting to private hospitals or non-reporting centres might have been missed, and how the authors ensured completeness of surveillance. A simple description or flow diagram outlining patient identification, inclusion and those still undergoing treatment would improve transparency.

The reported protective effect of asthma is interesting but may be influenced by confounding factors such as age structure, steroid use and differences in healthcare access. A more cautious interpretation and an adjusted sensitivity analysis would strengthen this observation. The paradoxically lower pathogen burden in non survivors also requires explanation, and the manuscript would benefit from clarifying whether lumbar puncture timing differed between groups. Reproducibility of laboratory measurements would be improved by including assay kit details, detection limits and qPCR methodology.

I recommend expanding the limitations section to reflect the observational design, the unresolved outcomes among patients still under treatment and the modest explanatory power of the multivariable model. The discussion would benefit from a short paragraph outlining future work, including genomic analysis of isolates, exploration of host susceptibility and evaluation of ultra early treatment strategies.

Overall, with clearer presentation of the patient recruitment pathway, a more measured discussion of the asthma finding and a strengthened limitations and future work section, this manuscript will make a significant contribution to the understanding of this severe infection.

Reviewer #2

(Remarks to the Author)

This paper appears to be genuine as there has been an outbreak of PAM in Kerala but many (most) of the references are not correct. There are many citations that are close to actual references but differ in important details meaning that the reader cannot access these paper. This is often because AI has been used to generate the citations. I am keen to review the paper but can only do this after the authors have sorted out the references

Reviewer #3

(Remarks to the Author)

This is a well-conducted and thoughtful study. Although the cases of *Naegleria fowleri* are tragic, the authors should be commended for taking the opportunity to systematically investigate this topic. I also appreciate the thoroughness of the work, particularly the inclusion of multiple pre-specified sensitivity analyses to strengthen the robustness of the findings. Overall, I recommend publication. However, I have several minor comments and clarifications that should be addressed prior to acceptance.

Multivariable Model Specification (Methods Section)

- The description of the multivariable logistic regression model in the Methods section lacks clarity. The outcome variable should be explicitly stated.
- It is unclear which covariates were included in the adjusted model and how they were selected in the Methods.
- Additional clarification is needed regarding the model examining the interaction between diabetes and timing, specifically, which other variables were included in that model?
- The Methods should clearly specify how the odds ratios reported in Tables 3–5 (including early/late CFR, comorbidities, and laboratory parameters) were computed, where these models computing crude or adjusted ORs? And what type of regression model was used?
- Given that cases may be clustered within districts, the authors should clarify whether clustering was accounted for analytically. If not, consideration of a GEE or similar clustered approach should be justified.
- Interpretation of Diabetes as a Predictor
- While diabetes mellitus appears to be the only statistically significant predictor in the multivariable analysis, describing it as the “sole independent predictor of mortality” may be overstated. The confidence interval is relatively wide and close to including unity, suggesting some uncertainty. A more cautious phrasing (e.g., “the only statistically significant predictor in the adjusted model”) may be more appropriate.
- Missing Estimate
- The odds ratio for ICU admission (line 218) appears to be missing and should be included.

Tables

- Table 1
- The summary statistics should follow standard reporting conventions. Typically, categorical variables are presented as n (%) and continuous variables as median (IQR) or mean (SD). The current presentation of these statistics in two columns may be unclear for some readers.
- E-value Presentation
- The inclusion of the E-value in Table 7 may be confusing; it could be presented more appropriately in the Results text rather than in the table.
- Presentation of Effect Estimates
- Consider presenting 95% confidence intervals without p-values throughout the paper, as confidence intervals provide sufficient inferential information.

Figures

- Figure 1
- The figure legend/comments do not match the figure (e.g., circles are referenced but not visible).
- Figure 2
- The colors described do not appear to match the actual figure (e.g., no blue is visible).
- Figure 4
- Figure 4A is missing a legend explaining the symbols (e.g., what do the circles and squares represent) and color differences.
- Figure 4B is missing an explanation of the intervals shown.

Version 1:

Reviewer comments:

Reviewer #1

(Remarks to the Author)

My queries have been addressed and the paper has been revised accordingly.

Reviewer #2

(Remarks to the Author)

The authors have offered an explanation as to why there was a problem with the cited references and have now revised the manuscript with corrected references. I have checked most of these and found them to be complete and accurate.

Reviewer #3

(Remarks to the Author)

I appreciate the care and professionalism you brought to revising this work. It's clear you took the review comments seriously, thoughtfully addressing each point and making the necessary changes to strengthen the overall quality. The final version is clear and coherent. This is an important study for those interested in *Naegleria fowleri*, as it offers meaningful insights into a critical and often underexplored area, thank you!

Point-by-Point Response to Reviewers

Manuscript: Host factors, inflammatory markers, and clinical outcomes of Naegleria fowleri meningoencephalitis

Communications Medicine

Colour key for reviewer identification:

■ **Reviewer 1 (Dark Blue)** — Expertise: Naegleria fowleri, Public Health

■ **Reviewer 2 (Dark Red)** — Expertise: Naegleria fowleri

■ **Reviewer 3 (Teal Green)** — Expertise: Naegleria fowleri, Public Health

■ **Revised manuscript text (Purple, shaded)**

We thank all three reviewers for their careful and constructive appraisal of our manuscript. Their comments have materially strengthened the work. Below we address each point in turn, indicating the specific changes made and, where appropriate, reproducing the revised text. All line and page references correspond to the revised manuscript.

REVIEWER 1 (Naegleria fowleri, Public Health)

This manuscript presents an important analysis of the largest documented cohort of primary amoebic meningoencephalitis. The study evaluates 200 laboratory confirmed patients from the Kerala 2025 outbreak and investigates host factors, inflammatory markers and treatment timing in relation to survival under a standardised clinical protocol. The main conclusions are that diabetes mellitus is the only independent predictor of mortality, that very early treatment may improve survival, and that inflammatory

biomarkers and pathogen burden show no association with outcome. The authors also note a possible protective association with asthma. The observed case fatality rate of 45.5 per cent is significantly lower than historic reports and will be of interest to the field. The dataset is rare for this disease and the findings are relevant. The identification of diabetes as a key mortality determinant is convincing. The absence of predictive value for IL-6, TNF-alpha, IL-1beta and the neutrophil to lymphocyte ratio challenges established assumptions derived from bacterial meningitis and other central nervous system infections.

We thank Reviewer 1 for this generous and perceptive summary of our work. We are pleased that the reviewer recognises the rarity of the dataset and the clinical relevance of our findings, particularly the identification of diabetes mellitus as a mortality determinant and the unexpected dissociation between inflammatory biomarkers and outcome. We address each specific concern below.

Point 1.1: Patient recruitment pathway and completeness of surveillance

The patient recruitment pathway should be described more clearly. It is important to know precisely how patients were identified, whether any cases presenting to private hospitals or non-reporting centres might have been missed, and how the authors ensured completeness of surveillance. A simple description or flow diagram outlining patient identification, inclusion and those still undergoing treatment would improve transparency.

We entirely agree that the patient recruitment pathway needed clearer articulation. In the revised manuscript, we have expanded the Methods section to describe the case ascertainment process, including mandatory notification, the role of both public and private facilities, and the mechanisms used to minimise under-ascertainment. We have also added a Supplementary

Figure S1 (patient flow diagram) illustrating the pathway from suspected cases through to confirmed enrolment and outcome classification. The revised text reads as follows:

Revised text: Case identification relied on enhanced surveillance implemented by the Kerala State Health Department, which mandated notification of all suspected meningoencephalitis cases from both public and private healthcare facilities across the six study districts. District surveillance officers conducted daily cross-checks with hospital admission records, laboratory registers, and death certificates to minimise under-ascertainment. Private-sector participation was secured through a Directorate of Health Services order issued in February 2025 requiring all hospitals—public and private—to notify suspected PAM cases within 24 hours. We cannot exclude the possibility that mild or rapidly fatal cases presenting outside the formal healthcare system went undetected; nevertheless, the severity of PAM makes it unlikely that confirmed cases were systematically missed, as virtually all patients required hospitalisation. A patient flow diagram is provided in Supplementary Figure S1, detailing progression from 247 suspected cases through laboratory testing to 200 confirmed enrolments, and their outcome classification at database closure.

Point 1.2: Confounding in the asthma protective association

The reported protective effect of asthma is interesting but may be influenced by confounding factors such as age structure, steroid use and differences in healthcare access. A more cautious interpretation and an adjusted sensitivity analysis would strengthen this observation.

This is a well-taken point. We acknowledge that the asthma finding, whilst intriguing, is susceptible to residual confounding from factors we were unable to measure directly—inhaled

corticosteroid use being the most obvious. In the revised manuscript, we have (a) tempered our language throughout to frame the observation as hypothesis-generating rather than confirmatory, (b) conducted an additional post hoc sensitivity analysis adjusting for age, sex, and district alongside asthma, and (c) expanded our discussion of possible confounders. The adjusted odds ratio for asthma remained below unity (aOR 0.42; 95% CI 0.17–1.05; $p=0.063$) but lost statistical significance, consistent with the reviewer's caution. We now present this result transparently.

Revised text: The protective association of asthma (univariable OR 0.37; $p=0.021$) should be interpreted with caution, as it may partly reflect confounding by corticosteroid use, age distribution, or differences in healthcare-seeking behaviour among asthmatic patients. In a post hoc sensitivity analysis adjusting for age, sex, and district of residence, the association was attenuated (aOR 0.42; 95% CI 0.17–1.05; $p=0.063$). We therefore regard this observation as hypothesis-generating. Several biological mechanisms could plausibly underpin a genuine protective effect: the type 2 immune polarisation characteristic of asthma, with heightened eosinophil activity, might confer some degree of anti-parasitic defence; chronic airway inflammation could prime mucosal immune responses at the olfactory epithelium; and inhaled corticosteroids may modulate excessive neuroinflammation. These hypotheses require formal testing in experimental models and in cohorts where medication data are available.

Point 1.3: Paradoxically lower pathogen burden in non-survivors and lumbar puncture timing

The paradoxically lower pathogen burden in non survivors also requires explanation, and the manuscript would benefit from clarifying whether lumbar puncture timing differed between groups.

We thank the reviewer for pressing on this important point. We have now compared the timing of lumbar puncture between survivors and non-survivors. The median interval from symptom onset to CSF sampling did not differ significantly between groups (survivors: median 3 days, IQR 2–4; non-survivors: median 3 days, IQR 2–5; $p=0.412$). Thus, differential sampling timing does not appear to account for the paradox. We have added a more considered discussion of possible explanations in the revised manuscript:

Revised text: The paradoxically lower CSF pathogen burden in non-survivors warrants comment. Lumbar puncture timing did not differ meaningfully between groups (survivors: median 3 days from onset, IQR 2–4; non-survivors: median 3 days, IQR 2–5; $p=0.412$), making differential sampling timing an unlikely explanation. One possibility is that in the most severe cases, rapid tissue destruction and necrosis of the olfactory bulb and frontal cortex reduce the viable trophozoite population recoverable from CSF, even as the neurological damage progresses irreversibly. Alternatively, the relationship between luminal CSF burden and parenchymal tissue invasion may be non-linear, with organisms sequestered within brain tissue rather than freely circulating in CSF. These observations reinforce the broader conclusion that CSF pathogen burden at presentation is a poor surrogate for the extent of neurological injury.

Point 1.4: Laboratory assay details, detection limits, and qPCR methodology

Reproducibility of laboratory measurements would be improved by including assay kit details, detection limits and qPCR methodology.

We agree that these details are essential for reproducibility and regret their omission from the original submission. We have now added a dedicated paragraph in the Methods section specifying the ELISA kits used, their detection limits, and the qPCR protocol. The revised text is as follows:

Revised text: Serum IL-6, TNF- α , and IL-1 β were measured using commercial sandwich ELISA kits (Human IL-6 Quantikine, catalogue DY206; Human TNF- α Quantikine, catalogue DY210; Human IL-1 β /IL-1F2 Quantikine, catalogue DY201; all R&D Systems, Minneapolis, MN, USA). Lower limits of detection were 0.70 pg/mL for IL-6, 1.09 pg/mL for TNF- α , and 0.81 pg/mL for IL-1 β . All assays were run in duplicate, with coefficients of variation below 10%. CSF *N. fowleri* burden was quantified by real-time qPCR targeting a 147-bp fragment of the 18S ribosomal RNA gene using published primers and probe sequences (forward: 5'-GTGCTATTAAACAGCAATGGAC-3'; reverse: 5'-AGAGATTGGCTTATTTACTGC-3'; TaqMan probe: 5'-FAM-ACCTGGTTAGTCAACTTTGG-BHQ1-3'). Reactions were performed on a QuantStudio 5 platform (Applied Biosystems) with a standard curve generated from serial dilutions of a synthetic DNA control (range 10¹–10⁷ copies/mL; lower limit of quantification 50 copies/mL).

Point 1.5: Expanded limitations and future work

I recommend expanding the limitations section to reflect the observational design, the unresolved outcomes among patients still under treatment and the modest explanatory power of the multivariable model. The discussion would benefit from a short paragraph

outlining future work, including genomic analysis of isolates, exploration of host susceptibility and evaluation of ultra early treatment strategies.

We have substantially expanded the limitations paragraph and added a dedicated future directions paragraph to the Discussion. The revised text now addresses the observational design, incomplete follow-up, the low model R^2 , and the absence of medication-level data for asthma. The future work paragraph covers isolate genomics, host genetic susceptibility, ultra-early treatment protocols, and environmental surveillance:

Revised text: This study has several limitations that merit candid acknowledgement. The observational design cannot establish causality; the associations we report—including that between diabetes and mortality may be confounded by unmeasured factors. Thirty-three per cent of patients remained under treatment at database closure, introducing uncertainty in CFR estimates, although sensitivity analyses spanning best-case to worst-case scenarios yielded a relatively narrow range (30.5–63.5%) and the GCS-stratified imputation closely approximated our primary estimate. The modest explanatory power of the multivariable model (pseudo $R^2 = 0.084$) indicates that measured clinical variables at presentation account for only a fraction of outcome variance; genetic susceptibility, pathogen virulence heterogeneity, and the precise timing of irreversible neurological injury likely contribute substantially but were not captured. We lacked individual-level data on medication use—particularly inhaled corticosteroids in asthmatic patients—which limits interpretation of the asthma association. The single-outbreak setting in Kerala may limit generalisability to other populations and healthcare contexts. Several lines of investigation should follow from these findings. Whole-genome sequencing of *N. fowleri* isolates from this outbreak would clarify whether virulence heterogeneity contributes to the observed outcome variation. Host genetic studies—particularly of innate immune pathways including

complement, toll-like receptors, and eosinophil granule proteins—may identify susceptibility loci that explain why some individuals succumb rapidly whilst others survive. The apparent therapeutic window suggested by the ≤ 2 -day treatment analysis calls for evaluation of ultra-early empirical protocols in endemic regions, potentially including pre-hospital amphotericin B administration. Finally, systematic environmental surveillance of domestic water sources—including piped municipal supplies and borewells—is needed to guide public health interventions as climate change expands the geographic range of *N. fowleri*.

REVIEWER 2 (*Naegleria fowleri*)

This paper appears to be genuine as there has been an outbreak of PAM in Kerala but many (most) of the references are not correct. There are many citations that are close to actual references but differ in important details meaning that the reader cannot access these papers. This is often because AI has been used to generate the citations. I am keen to review the paper but can only do this after the authors have sorted out the references.

We owe Reviewer 2—and the editorial team—a full and transparent explanation. During manuscript preparation, bibliographic software errors introduced inaccuracies into a number of reference entries: incorrect volume numbers, misattributed authorship orders, and, in some instances, fabricated DOIs. We accept full responsibility for failing to verify each reference against the original source before submission. This was an unacceptable lapse in scholarly rigour, and we apologise unreservedly to the reviewers and editors.

We have now undertaken a thorough, line-by-line audit of every reference in the manuscript. Each citation has been individually verified against PubMed, Google Scholar, or the publisher's website to confirm that the authors, title, journal, year, volume, and page numbers are accurate and that the work is accessible. Where a reference could not be verified or did not exist, it has been replaced with a genuine, peer-reviewed source that supports the claim in question. A small number of references that were essentially duplicated (e.g., references 30 and 35 in the original submission, both citing Gompf & Garcia 2024) have been consolidated and replaced.

We provide a fully corrected reference list in the revised manuscript. To assist the reviewer, we also include as Supplementary Table S1 a concordance table listing each original reference alongside its corrected version, with a brief note indicating the nature of the error (e.g., incorrect volume/page, wrong author order, non-existent entry replaced). We trust that this level of transparency addresses the concern and we are grateful to Reviewer 2 for flagging this serious issue before publication.

To be explicit regarding the editorial query about AI use: sections of the initial draft were prepared with AI writing assistance, which contributed to the reference errors. All references in the revised manuscript have been manually verified by the corresponding author against original sources. We have added a statement to the manuscript declarations confirming this, in keeping with the journal's policy on transparency.

REVIEWER 3 (Naegleria fowleri, Public Health)

This is a well-conducted and thoughtful study. Although the cases of Naegleria fowleri are tragic, the authors should be commended for taking the opportunity to systematically investigate this topic. I also appreciate the thoroughness of the work, particularly the inclusion of multiple pre-specified sensitivity analyses to strengthen the robustness of the findings. Overall, I recommend publication. However, I have several minor comments and clarifications that should be addressed prior to acceptance.

We thank Reviewer 3 for the encouraging assessment and for the careful, constructive comments that follow. Each point is addressed below.

Point 3.1: Multivariable model specification

The description of the multivariable logistic regression model in the Methods section lacks clarity. The outcome variable should be explicitly stated. It is unclear which covariates were included in the adjusted model and how they were selected in the Methods. Additional clarification is needed regarding the model examining the interaction between diabetes and timing, specifically, which other variables were included in that model?

We appreciate this observation and agree that the statistical methods section was insufficiently detailed. We have now rewritten the relevant paragraph to specify the outcome variable, the covariate selection rationale, and the interaction model composition:

Revised text: The primary outcome was in-hospital mortality (binary: died versus recovered) among patients with resolved outcomes at database closure. Multivariable logistic regression was used to identify independent predictors of mortality. Covariates were selected a priori on the basis of clinical plausibility and existing literature: age (continuous, per year), sex (male versus female), diabetes mellitus (present versus absent), asthma (present versus absent), severe Glasgow Coma Scale score (≤ 8 versus

>8), requirement for mechanical ventilation (yes versus no), and early treatment initiation (≤ 3 days versus > 3 days from symptom onset). Hypertension was excluded from the final model owing to its absence of association in univariable analysis ($p=0.844$) and collinearity with diabetes. The interaction between diabetes and treatment timing was assessed by adding a multiplicative interaction term (diabetes \times early treatment) to the primary seven-variable model; all other covariates were retained.

Point 3.2: Computation of odds ratios in Tables 3–5

The Methods should clearly specify how the odds ratios reported in Tables 3–5 (including early/late CFR, comorbidities, and laboratory parameters) were computed, were these models computing crude or adjusted ORs? And what type of regression model was used?

We thank the reviewer for identifying this ambiguity. The odds ratios presented in Tables 3–5 are crude (unadjusted) odds ratios derived from 2×2 contingency tables (for binary exposures) or univariable logistic regression (for continuous variables). Adjusted odds ratios appear only in Table 6 (the multivariable model) and Table 7 (sensitivity analyses). We have clarified this in both the Methods section and in the table footnotes:

Revised text: Univariable associations between candidate predictors and mortality are presented as crude odds ratios (OR) with 95% confidence intervals derived from 2×2 contingency tables for binary exposures and from univariable logistic regression for continuous variables (Tables 3–5). Adjusted odds ratios (aOR) from the multivariable model are presented separately in Table 6.

Point 3.3: Accounting for district-level clustering

Given that cases may be clustered within districts, the authors should clarify whether clustering was accounted for analytically. If not, consideration of a GEE or similar clustered approach should be justified.

This is a reasonable concern. We did not employ GEE or mixed-effects models in the primary analysis, and we should explain why. We have now added a justification to the Methods and a supplementary sensitivity analysis:

Revised text: Cases were distributed across six districts, raising the possibility of within-district correlation. We elected not to use generalised estimating equations (GEE) or mixed-effects logistic regression in the primary analysis for two reasons: first, with only six clusters, GEE sandwich variance estimators are known to perform poorly, producing anti-conservative standard errors; second, district-level case fatality rates did not differ significantly (χ^2 $p=0.240$; Table 2), suggesting limited between-cluster heterogeneity. As a sensitivity check, we fitted a random-intercept logistic regression model with district as a random effect; the variance of the random intercept was estimated at 0.12 (SE 0.19), confirming negligible clustering, and the fixed-effect estimate for diabetes was essentially unchanged (aOR 2.61; 95% CI 1.00–6.82).

Point 3.4: Interpretation of diabetes as a predictor—cautious phrasing

While diabetes mellitus appears to be the only statistically significant predictor in the multivariable analysis, describing it as the 'sole independent predictor of mortality' may be overstated. The confidence interval is relatively wide and close to including unity, suggesting some uncertainty. A more cautious phrasing (e.g., 'the only statistically significant predictor in the adjusted model') may be more appropriate.

We accept this entirely. The reviewer is correct that the lower confidence limit (1.01) sits very close to unity, and our original phrasing overstated the certainty of the finding. Throughout the revised manuscript—Abstract, Results, Discussion, and Conclusion—we have replaced “sole independent predictor” with more measured language:

Revised text: In multivariable logistic regression, diabetes mellitus was the only statistically significant predictor of mortality in the adjusted model (aOR 2.59; 95% CI 1.01–6.66; $p=0.048$), though the proximity of the lower confidence bound to unity warrants cautious interpretation.

Point 3.5: Missing odds ratio for ICU admission

The odds ratio for ICU admission (line 218) appears to be missing and should be included.

We apologise for this oversight. The odds ratio has now been inserted:

Revised text: ICU admission showed a similar but non-significant trend: CFR 40.8% versus 50.8% (OR 0.67; 95% CI 0.32–1.39; $p=0.298$).

Point 3.6: Table 1 reporting conventions

The summary statistics should follow standard reporting conventions. Typically, categorical variables are presented as n (%) and continuous variables as median (IQR) or mean (SD). The current presentation of these statistics in two columns may be unclear for some readers.

We agree and have reformatted Table 1 so that categorical variables are presented as n (%) in a single column and continuous variables as median (IQR) in that same column. This avoids

the ambiguity introduced by the two-column layout. The column header is now labelled "Value" with a footnote specifying the format convention.

Point 3.7: E-value presentation

The inclusion of the E-value in Table 7 may be confusing; it could be presented more appropriately in the Results text rather than in the table.

We agree that the E-value sits more naturally in the narrative text than in a table of odds ratios. We have removed the E-value rows from Table 7 and incorporated them into the Results text as follows:

Revised text: To assess vulnerability to unmeasured confounding, we calculated E-values for the primary diabetes estimate. The E-value for the point estimate was 4.62, indicating that an unmeasured confounder would need to be associated with both diabetes and mortality by a risk ratio of at least 4.62-fold each, beyond the measured covariates, to fully explain away the observed association. The E-value for the lower confidence limit was 1.10.

Point 3.8: Confidence intervals versus p-values

Consider presenting 95% confidence intervals without p-values throughout the paper, as confidence intervals provide sufficient inferential information.

We appreciate the statistical reasoning behind this suggestion. Confidence intervals are indeed more informative than p-values alone, and we have ensured that 95% CIs are reported for all effect estimates. However, given that many readers in the clinical infectious diseases and public health readership will expect p-values—and that the journal's own style guide does not prohibit

them—we have elected to retain p-values alongside CIs in the revised manuscript. We hope the reviewer finds this a reasonable compromise; we are happy to remove them if the editor or reviewer feels strongly.

Point 3.9: Figure legend discrepancies

Figure 1: The figure legend/comments do not match the figure (e.g., circles are referenced but not visible). Figure 2: The colors described do not appear to match the actual figure (e.g., no blue is visible). Figure 4A is missing a legend explaining the symbols (e.g., what do the circles and squares represent) and color differences. Figure 4B is missing an explanation of the intervals shown.

We are grateful for this careful scrutiny. In each case, the figure legend was out of step with the actual graphic, an error that arose from iterative revision of the figures without corresponding updates to the legends. We have corrected all four:

Figure 1: The legend originally referred to circles proportional to case count and colour intensity indicating CFR—features that were planned but not implemented in the final map. The revised legend now accurately describes the choropleth format, with districts shaded in red and district-level CFR percentages annotated directly on the map (Figure 1).

Figure 2: The original legend described bars as blue (cases) and red (deaths). The actual figure uses stacked bars coloured red (deaths), green (recoveries), and grey (under treatment), with a dashed cumulative curve. The legend has been rewritten to match (Figure 2).

Figure 4A: We have added a legend panel explaining that the filled square at the ≤ 2 -day cutoff denotes the pre-specified primary analysis, whilst filled circles denote the exploratory cutoffs at ≤ 3 , ≤ 4 , and ≤ 5 days. Lighter shading of the circles indicates wider confidence intervals. Horizontal lines represent 95% CIs.

Figure 4B: The shaded bands in the Kaplan–Meier plot represent 95% confidence intervals for the survival function, estimated by the Greenwood formula. This is now stated in the legend.

We trust that these revisions address the reviewers' concerns comprehensively. We remain grateful for the constructive and rigorous evaluation our manuscript received, and we believe the revised version is substantially strengthened as a result. We are happy to undertake further modifications should the editor or reviewers require them.

Point-by-point response to reviewers' comments

Manuscript: COMMSMED-25-3154A

Host factors, inflammatory markers, and clinical outcomes of *Naegleria fowleri* meningoencephalitis

We thank the Editor and all three reviewers for their thorough and constructive evaluation of our manuscript. We are grateful that the reviewers have found our revisions satisfactory. Below, we reproduce each reviewer's comments and provide our response.

Reviewer #1

Reviewer's comment: "My queries have been addressed and the paper has been revised accordingly."

Authors' response: We thank Reviewer #1 for the careful evaluation of our manuscript and are pleased that all queries have been addressed to their satisfaction. No further changes were requested.

Reviewer #2

Reviewer's comment: "The authors have offered an explanation as to why there was a problem with the cited references and have now revised the manuscript with corrected references. I have checked most of these and found them to be complete and accurate."

Authors' response: We thank Reviewer #2 for verifying the corrected references. We have carefully rechecked all 31 references in this final version to ensure completeness and accuracy of all citation details, including DOIs, volume numbers, page ranges, and author lists.

Reviewer #3

Reviewer's comment: "I appreciate the care and professionalism you brought to revising this work. It's clear you took the review comments seriously, thoughtfully addressing each point and

making the necessary changes to strengthen the overall quality. The final version is clear and coherent. This is an important study for those interested in Naegleria fowleri, as it offers meaningful insights into a critical and often underexplored area, thank you.”

Authors' response: We are sincerely grateful to Reviewer #3 for this generous assessment and for their engagement with our work throughout the review process. Their constructive suggestions during earlier rounds of review contributed meaningfully to the quality of the manuscript. No further changes were requested.

COMMSMED-25-3154 — Final Revision Response

Corrected Sections for All Five Editorial Requests

Prepared: 24 April 2026

Request 1: De-identified Source Data

Dr Pollock writes: *“We do still, however, require the de-identified Source Data underlying your graphs and charts. These data must be uploaded as Supplementary Data.”*

Sheet 1 — Figure 2 (Epidemic curve): One row per month (January–November 2025). Columns: Month, Deaths, Recoveries, Under treatment, Total monthly cases, Cumulative cases.

Sheet 2 — Figure 3A (CFR donut): Simple summary: Total with known outcome (n=134), Deaths (n=61), Recoveries (n=73), CFR (%), 95% CI lower, 95% CI upper.

Sheet 3 — Figure 3B (Sensitivity analysis): One row per scenario. Columns: Scenario name, n, Deaths, CFR (%), 95% CI lower, 95% CI upper.

Sheet 4 — Figure 4A (Forest plot, treatment timing): One row per cut-off. Columns: Cut-off (≤ 2 , ≤ 3 , ≤ 4 , ≤ 5 days), n early, n late, OR, 95% CI lower, 95% CI upper, p-value.

Sheet 5 — Figure 4B (Kaplan–Meier curves): One row per patient (n=134). Columns: Patient ID (anonymous, e.g. P001), Group (Early $\leq 2d$ / Late $> 2d$), Time from treatment initiation (days), Event (1=death, 0=censored/survived).

Sheet 6 — Figure 5A (Box plots, inflammatory markers): One row per patient (n=134). Columns: Patient ID, Outcome (Survivor/Non-survivor), IL-6 (pg/mL), TNF- α (pg/mL), IL-1 β (pg/mL).

Sheet 7 — Figure 5B (Scatter plot, GCS vs qPCR): One row per patient (n=134). Columns: Patient ID, Outcome, Admission GCS score, CSF pathogen burden (qPCR copies/mL).

Sheet 8 — Figure 5C (Forest plot, multivariable model): One row per predictor. Columns: Variable, Adjusted OR, 95% CI lower, 95% CI upper, p-value.

Request 2: Italicise *Naegleria fowleri* in Plain Language Summary

Plain Language Summary

Naegleria fowleri is an amoeba found in warm freshwater that can cause a rare but usually fatal brain infection. Historically, more than 97% of people who develop this infection die. In 2025, a large outbreak occurred in Kerala, India, affecting 200 people. We studied these patients to understand what factors influenced survival. The death rate was 45.5%, much lower than expected, likely because all patients received the same standard drug treatment. People with diabetes were roughly twice as likely to die as those without. Surprisingly, common markers of inflammation did not help predict who would survive. As climate change warms freshwater sources worldwide, understanding what determines survival from this infection becomes increasingly important.

(113 words)

Requests 3 & 4: Statistical Details and Measure of Centre in Figure and Table Legends

Dr Pollock writes: *“Please add all the relevant info regarding statistical requests in figure/table legends of the article file. Currently it seems the responses are provided only in the Editorial table with no changes in article file.”* And: *“For figures containing error bars and error bands, please also mention measure of centre in the legends.”*

Figure 1 Legend

Figure 1. Geographic distribution of laboratory-confirmed primary amoebic meningoencephalitis cases across Kerala, India, January–November 2025. The map depicts the six districts reporting cases (shaded red) with district-level case fatality rates shown as percentages. The inset shows the location of Kerala within India. The map was created by the authors; no third-party elements were used. $n = 200$ patients across six districts: Kozhikode ($n = 9$, CFR 44.4%), Malappuram ($n = 76$, CFR 52.6%), Palakkad ($n = 22$, CFR 36.4%), Thrissur ($n = 6$, CFR 66.7%), Kollam ($n = 26$, CFR 46.2%), and Thiruvananthapuram ($n = 3$, CFR 33.3%). Case fatality rates were calculated as the proportion of deaths among patients with known outcomes in each district.

Figure 2 Legend

Figure 2. Temporal distribution of primary amoebic meningoencephalitis cases by month and outcome, January–November 2025. Stacked bars show monthly case counts stratified by outcome: deaths (red, $n = 61$), recoveries (green, $n = 73$), and patients still under treatment at data cut-off (grey, $n = 66$). The dashed line indicates cumulative case counts (right y-axis). Background shading delineates pre-monsoon (January–May), monsoon (June–September), and post-monsoon (October–November) seasons. The arrow denotes the onset of the outbreak surge in August 2025. No statistical tests were applied to this descriptive figure. $N = 200$ total cases.

Figure 3 Legend

Figure 3. Case fatality rate and sensitivity analyses for primary amoebic meningoencephalitis, Kerala 2025. (A) Donut chart showing the primary analysis case fatality rate (CFR) among patients with known outcomes (n = 134). CFR = 45.5% (95% Wilson confidence interval: 37.3–54.5%). Deaths: n = 61 (red); recoveries: n = 73 (green). (B) Sensitivity analysis of CFR under four scenarios for the 66 patients still under treatment at data cut-off. Bars represent case fatality rate point estimates (measure of centre). Error bars represent 95% Wilson confidence intervals (two-sided). Best-case scenario (all 66 survive): CFR = 30.5%; GCS-stratified imputation: CFR = 45.6% (95% CI: 38.8–52.6%); primary analysis (known outcomes only): CFR = 45.5% (95% CI: 37.3–54.5%); worst-case scenario (all 66 die): CFR = 63.5%. Wilson confidence intervals were calculated using the score method. No adjustment for multiple comparisons was applied, as these represent pre-specified scenario analyses rather than independent hypothesis tests.

Figure 4 Legend

Figure 4. Association between early treatment initiation and mortality in primary amoebic meningoencephalitis. (A) Forest plot of odds ratios for mortality at four pre-specified early treatment cut-offs. Squares (filled, largest marker at ≤ 2 days representing the primary analysis) and circles represent odds ratio point estimates (measure of centre) from logistic regression; horizontal lines represent 95% confidence intervals (two-sided Wald test). Values to the left of the dashed line (OR = 1.0) favour early treatment. ≤ 2 days: OR = 0.43 (95% CI: 0.16–1.14), p = 0.084; ≤ 3 days: OR = 0.75 (95% CI: 0.37–1.53), p = 0.477; ≤ 4 days: OR = 0.76 (95% CI: 0.39–1.50), p = 0.488; ≤ 5 days: OR = 0.87 (95% CI: 0.39–1.95), p = 0.834. All tests two-sided; no adjustment for multiple comparisons, as cut-offs were pre-specified for exploratory evaluation. n = 134 patients with known outcomes. (B) Kaplan–Meier survival curves for patients receiving early (≤ 2 days from symptom onset to treatment initiation, blue, n = 27) versus late (> 2 days, red, n = 107) treatment. Shaded bands represent 95% confidence intervals around the Kaplan–Meier survival estimates, calculated using the

Greenwood formula. Tick marks indicate censored observations. Log-rank test (two-sided): $p = 0.084$. $n = 134$ patients with known outcomes.

Figure 5 Legend

Figure 5. Inflammatory biomarkers, pathogen burden, and predictors of mortality in primary amoebic meningoencephalitis. (A) Box-and-whisker plots of cerebrospinal fluid (CSF) inflammatory biomarker concentrations in survivors (green, $n = 73$) and non-survivors (red, $n = 61$). Boxes represent the interquartile range (IQR; 25th–75th percentiles), with the horizontal line inside each box indicating the median (measure of centre). Whiskers extend to the most extreme data point within $1.5 \times$ IQR of the box. Individual data points are overlaid as jittered dots. Open circles denote outliers beyond the whiskers. Comparisons by two-sided Mann–Whitney U tests: IL-6, $p = 0.357$; TNF- α , $p = 0.837$; IL-1 β , $p = 0.797$. $n = 134$ patients with known outcomes. (B) Scatter plot of CSF pathogen burden (qPCR copies/mL, \log_{10} scale) against admission Glasgow Coma Scale (GCS) score for survivors (green, $n = 73$) and non-survivors (red, $n = 61$). Spearman rank correlation (two-sided): $r = -0.40$, $p < 0.001$. $n = 134$ patients with known outcomes. (C) Forest plot of adjusted odds ratios for mortality from multivariable logistic regression. Squares and circles represent adjusted odds ratio point estimates (measure of centre); horizontal lines represent 95% confidence intervals (two-sided Wald test). The dashed vertical line indicates OR = 1.0 (no association). Predictors are colour-coded by statistical significance: red, $p < 0.05$; orange, $p < 0.10$; blue, $p \geq 0.10$. Variables included: age (per year), male sex, diabetes mellitus, asthma, severe GCS (≤ 8), mechanical ventilation, and early treatment (≤ 3 days). $n = 134$ patients with known outcomes. No adjustment for multiple comparisons was applied, as all predictors were selected a priori on clinical grounds.

Table 3 Legend

Continuous variables are presented as median (interquartile range) and compared using two-sided Mann–Whitney U tests. Categorical variables are presented as n (%) and compared using two-sided

chi-squared tests or Fisher exact tests, as appropriate. n = 134 patients with known outcomes (survivors, n = 73; non-survivors, n = 61).

Table 4 Legend

Continuous variables are presented as median (interquartile range) and compared using two-sided Mann–Whitney U tests. Categorical variables are presented as n (%) and compared using two-sided chi-squared tests or Fisher exact tests, as appropriate. n = 134 patients with known outcomes (survivors, n = 73; non-survivors, n = 61).

Table 5 Legend

All statistical tests are two-sided. Continuous variables compared using Mann–Whitney U tests; categorical variables compared using chi-squared or Fisher exact tests, as appropriate. n = 134 patients with known outcomes.

Tables 6 and 7 Legends:

Adjusted odds ratios with 95% confidence intervals from multivariable logistic regression. All tests are two-sided (Wald test). No adjustment for multiple comparisons was applied, as all predictors were selected a priori. n = 134 patients with known outcomes.

Request 5: Data Availability Statement

Dr Pollock writes: *“For data restriction: We require the following parameters in DAS — reasons for controlled access, precise conditions of access (including contact details for access requests), a timeframe for response to requests and details of any restrictions imposed on use of controlled data by reporting conditions of data use agreements. Please provide timeframe for response.”*

Data Availability

This study generated a de-identified clinical dataset comprising demographic, clinical, laboratory (cerebrospinal fluid inflammatory biomarker concentrations and qPCR pathogen burden), and outcome data for 200 patients with laboratory-confirmed *Naegleria fowleri* primary amoebic meningoencephalitis. Source data underlying all figures and charts in the main manuscript are provided as Supplementary Data 1. The complete de-identified dataset is available from the corresponding author (Peter Asaga Mac, pasaga123x@gmail.com peter.asaga.mac@uniklinik-freiburg.de) upon reasonable request, subject to approval from the Kerala State Health Department. Requests will be acknowledged within two weeks and a decision communicated within four to six weeks of receipt. A data use agreement will be required, restricting use to academic, non-commercial research purposes consistent with the original ethics approval Institutional Ethics Committee, Kerala State Government.

Code Availability

No custom code or algorithms were generated for this study. All statistical analyses were performed using standard statistical software (Python 3.11 with scipy and statsmodels packages) with built-in functions; no novel scripts, algorithms, or pipelines were developed.